# Modulatory Effect of Guinep (*Melicoccus bijugatus* Jacq) Fruit Pulp Extract on Isoproterenol-Induced Myocardial Damage in Rats. Identification of Major Metabolites Using High Resolution UHPLC Q-Orbitrap Mass Spectrometry

**DOI:** 10.3390/molecules24020235

**Published:** 2019-01-10

**Authors:** Chukwuemeka R. Nwokocha, Isheba Warren, Javier Palacios, Mario Simirgiotis, Magdalene Nwokocha, Sharon Harrison, Rory Thompson, Adrian Paredes, Jorge Bórquez, Astrid Lavado, Fredi Cifuentes

**Affiliations:** 1Department of Basic Medical Sciences Physiology Section, Faculty of Medical Sciences, The University of the West Indies, Mona, Kingston 7, KGN, Jamaica; Warren-Isheba.R@hotmail.com; 2Facultad Ciencias de la Salud, Instituto de EtnoFarmacología (IDE), Universidad Arturo Prat, Iquique 1110939, Chile; 3Instituto de Farmacia, Facultad de Ciencias, Universidad Austral de Chile, Valdivia 5110566, Chile; mario.simirgiotis@gmail.com; 4Department of Pathology, Faculty of Medical Sciences, University of the West Indies, Mona Campus, Kingston 7, KGN, Jamaica; magdanwokocha@yahoo.com (M.N.); sharon.harrison@uwimona.edu.jm (S.H.); rorykthompson@gmail.com (R.T.); 5Laboratorio de Química Biológica, Instituto Antofagasta, Universidad de Antofagasta, Antofagasta 1270300, Chile; adrian.paredes@uantof.cl; 6Departamento de Química, Facultad de Ciencias Básicas, Universidad de Antofagasta, Antofagasta 1270300, Chile; jorge.borquez@uantof.cl; 7Laboratorio de Fisiología Experimental, Instituto Antofagasta, Universidad de Antofagasta, Antofagasta 1270300, Chile; astrid.lavado@uantof.cl (A.L.); fredi.cifuentes@uantof.cl (F.C.)

**Keywords:** *Melicoccus bijugatus*, isoproterenol, myocardial infarction, high-resolution orbitrap mass spectrometry, rat

## Abstract

Guinep is traditionally used in the management of cardiovascular ailments. This study aims to evaluate its medicinal constituents and effects in the management of myocardial injury in an experimental isoproterenol (ISO) rat model. Sprague-Dawley rats were randomly assigned to four groups: Group 1 was the control group; Group 2 received *M. bijugatus* extract (100 mg/Kg; MB) for six weeks; Group 3 was given ISO (85 mg/Kg) i.p. twice during a 24-hour period; and Group 4 was given ISO (85 mg/Kg) i.p. and MB extract (100 mg/Kg) for six weeks. The MB was administered orally by gavage, daily. The blood pressure of conscious animals was measured, while ECG was performed under anesthesia. Blood and serum were collected for biochemical and hematological analysis. The ISO group treated with MB showed a significant decrease (*p* < 0.001) in (SBP), diastolic (DBP), mean arterial (MAP) and heart rate (HR) compared to the ISO only group. Conversely, MB treated rats that were not induced with ISO displayed a significant decreases (*p* < 0.001) in SBP, DBP, MAP, and HR. ISO significantly elevated the ST segment (*p* < 0.001) and shortened the QTc interval (*p* < 0.05), which were recovered after treatment with 100 mg/Kg of MB. In addition, the results showed a significant decrease (*p* < 0.001) in the heart to body weight ratio of the ISO group treated with MB compared to the ISO only group. Furthermore, the extract normalized the hematological values depressed by the ISO while significantly elevating the platelet count. UHPLC high-resolution orbitrap mass spectrometry analysis results revealed the presence of several antioxidants like vitamin C and related compounds, phenolic acids, flavonoid, fatty acids (oxylipins), and terpene derivatives. The results of this study indicated that *Melicoccus bijugatus* did display some cardio-protective effects in relation to myocardial injury.

## 1. Introduction

Cardiovascular disease (CVD) like acute myocardial infarction (AMI) is one of the leading causes of death; causing prolonged ischemia of the heart muscle resulting in tissue death or infarction in the myocardium [1]. This results in edema, a reduction in cardiac output, abnormalities in cardiac rhythm and transmission blocks that can further impair cardiac function. A reduction in cardiac output and arterial pressure may stimulate baroreceptor reflexes that lead to the activation of compensatory mechanisms, such as those of the sympathetic nerves and the renin-angiotensin-aldosterone system [2], and to elevations in cardiac biomarkers [3].

ISO is a potent non-selective beta-adrenergic receptor (β1 and β2) agonist that causes severe stress to the myocardium, resulting in infarct-like necrosis of the heart muscle [4]. Its proposed mechanism of action is through the auto-oxidation and production of free cytotoxic radicals [5], as well as hyper-stimulation of beta adrenoceptors [6]. These actions lead to the peroxidation of the cellular membrane, a change in membrane permeability and possible derangement of calcium ion pathway signaling, hypertrophy and myocardial injury [7,8]. The net effect of these includes a fall in DBP and MAP while SBP may remain unchanged, rise or fall (depending on the dose). Similarly, cardiac output may increase because of the positive inotropic and chronotropic effects of the drug, due to a decrease in peripheral vascular resistance.

*Melicoccus bijugatus*, known colloquially in Jamaica as Guinep, is a minor member of the *Sapindaceae* family [9]. The therapeutic effects of these fruits, including the management of diarrhea, cardiovascular disease, asthma and constipation, and as an astringent [10], were attributed to the combination of phenolic compounds and sugars. The phenolic content of this fruit was previously reported [9,11]. In the seed embryo, flavonoids, epicatechin, catechin, epigallocatechin, B-type procyanidins, naringenin, naringenin derivatives, phloretin, phloridzin, quercetin, myricetin and resveratrol, were identified in high amounts. The pulp of the fruit contains phenolic acid derivatives such as coumaric and ferulic acid derivatives, and hydroxycinnamic and sinapic acid [10,12]. This study aims to scientifically examine the mechanism of action of *M. bijugatus* in the management of cardiovascular ailments like AMI via an experimental rat model.

## 2. Results

### 2.1. Blood Pressure Changes and Electrocardiogram (ECG)

Table 1 shows that the extract group displayed a significant decrease (*p* < 0.001) in the MAP, SBP, and DBP when compared to the control (normotensive rats). There was also a significant decrease in the MAP, SBP, and DBP of the ISO group treated with MB compared to the ISO only group (*p* < 0.001).

The MB group displayed a significantly decreased (*p* < 0.01) HR when compared to the normotensive animals (control). There was also a significant decrease (*p* < 0.001) in the HR of the ISO plus MB treatment group when compared to the ISO only treatment group (Figure 1, Table 1). In addition, there was an observable (24.5%) increase in the HR of the ISO only group when compared to the control group (normotensive). However, this increase was not statistically significant.

Although the PP significantly decreased in all groups,the causes in the MB group (*p* < 0.001), ISO group (*p* < 0.01) and ISO + MB group (*p* < 0.01) were different compared to that of the control. Table 1 shows that ISO significantly (*p* < 0.05) increased the DBP, did not increase the SBP, and decreased the PP, consistent with decreased left ventricular compliance and increased myocardial stiffness [13]. MB significantly decreased the blood pressure (MAP, SBP, DBP), HR, and PP, which is proportional to the stroke volume [14].

In electrocardiograms from rats and mice, the beginning of the T-wave merges with the end of the QRS complex without an isoelectric ST segment. The changes, seen in the ECG, that affected the frequency of the waves can be seen to affect the HR also. ISO significantly elevated the ST segment (*p* < 0.001; Figure 1 and Figure 2A) and shortened the QTc interval (*p* < 0.05; Figure 1 and Figure 2B), which recovered after treatment with 100 mg/Kg of MB. The *M. bijugatus* of the plant did not, *per se*, cause any change.

### 2.2. Hematological Parameters

There were significant differences (*p* > 0.05) in the hematological parameters of the treatment groups when compared to the control group (Table 2 and Table 3). ISO significantly reduced the white blood count (WBC; *p* < 0.05), red blood count (RBC; *p* < 0.01), hematocrit (HCT; *p* < 0.01), mean cell volume (MCV; *p* < 0.01) and mean cell hemoglobin values (MCH; *p* < 0.01). *M. bijugatus* extract normalized the hematological values (WBC, RBC, HCT, MCV and MCH) depressed by ISO, while significantly (*p* < 0.01) elevating the platelet count.

### 2.3. Histo-Morphological Analysis

The microscopic changes in the muscle fibers of the heart were limited to the group that was exposed to both ISO and MB. The maximum dimension of the myocardial infarction was of 1.1 mm and appeared to be in the central region of the left ventricular muscle wall, which is the area furthest away from both endocardium and epicardium and is most susceptible to infarction from vascular compromise (Figure 3C).

Longitudinal and/or transverse sections of the large caliber abdominal blood vessels, i.e., aorta and caudal vena cava revealed no changes in the intima, media or external layer. The adventitia was composed primarily of brown fat. No inflammation was appreciated (data not shown).

### 2.4. Ratio of Heart Weight to Body Weight

The data does not suggest that *M. bijugatus* is responsible for any significant changes in body weight that could not be accounted for in the natural growth pattern of the Sprague-Dawley rat. ISO caused an elevated heart weight/body weight ratio and treatment with MB (100 mg/Kg) caused a significant decrease (Figure 4).

### 2.5. Identification of the Compounds

Hyphenated UHPLC-MS experiments were employed for the identification of unknown compounds in the fruits of *M. bijugatus* since it provides high resolution and an accurate mass product ion spectra (Figure 5, Table 4 and Appendix A). Combining Q-orbitrap HRAM (high resolution accurate mass) full MS scans and MS^n^ experiments, all compounds were tentatively identified, including simple phenolic acids, terpenes, fatty acids, and one glycosylated flavonoid. As far as we know, some of the compounds, for this species, were reported for the first time. Below is a detailed explanation of the identification.

#### 2.5.1. Simple Organic Acids and Sugars

Peaks 1–5 and 8 were identified as simple organic acids such as vitamin C and sucrose. Peak 1, with an [M − H]^−^ ion at *m*/*z*: 191.01933, was identified as citric acid (C_6_H_7_O_7_^−^) and Peak 2 as the isomer isocitric acid [4]. Peak 3 was identified as saccharose (C_12_H_21_O_11_^−^) and Peak 4 as glucose (C_6_H_11_O_6_^−^). Sucrose and glucose were already reported as important constituents of this fruit [10,12]. Peak 5, with an [M − H]^−^ ion at *m*/*z*: 111.00787, was identified as furoic acid (C_3_H_3_O_3_^−^) and Peak 16, with an [M − H]^−^ ion at *m*/*z*: 293.17587, was assigned to the bioactive, cell permeable, 1,4-benzoquinone embelin derivative (C_17_H_26_O_4_^−^) [15].

#### 2.5.2. Flavonoids

Peak 12, with an [M − H]^−^ ion at *m*/*z*: 477.11670, and the MS2 fragment at *m*/*z*: 314.04370 were identified as Isorhamnetin-3-*O*-Glucoside (C_22_H_21_O_11_^−^) [16].

#### 2.5.3. Phenolic Compounds

Peak 6 was identified as salicilic acid glucoside (C_13_H_15_O_8_^−^), the parent ion 299.07712 delivered a diagnostic salicylic acid ion at *m*/*z*: 137.02440 [17], and Peak 7 was identified as the dicoumarin aflavarin (C_24_H_21_O_9_^−^). Peaks 9 and 10, with pseudomolecular ions at *m*/*z*: 325.09277 and dioagnostic MS2 fragments at *m*/*z*: 163.02910, 145.02870 and 117.03368, were identified as coumaric acid glucoside and coumaric acid galactoside, respectively (C_15_H_17_O_8_^−^). The presence of coumaric acid glucoside was already reported in this fruit pulp [10]. In the same manner, Peaks 11 and 13 were identified as the related compounds feruloyl glucoside and feruloyl galactoside (C_16_H_19_O_9_^−^) showing diagnostic MS2 ions at 147.04449 and 193.05056. Peak 14 was identified as chlorogenic acid: 3-*O*-caffeoylquinic acid (3-CQA, C_16_H_17_O_9_^−^), with a diagnostic quinic acid ion at 191.05608 [18].

#### 2.5.4. Fatty Acids

Two peaks were tentatively identified as fatty acids known as oxylipins [19,20]. Peak 8, with a pseudomolecular ion at *m*/*z*: 325.18443, was identified as a trihydroxy-octadecatrienoic acid (C_18_H_29_O_5_^−^) and Peak 2, with a pseudomolecular ion at *m*/*z*: 311.16876, was identified as a hydroxyheptadecatrienoic acid (C_17_H_27_O_7_^−^) [21].

#### 2.5.5. Terpenes and Related Compounds

Peak 17 was identified as the lactone sedanenolide (C_12_H_12_O_2_^−^) [22] and Peak 18 was assigned to the sesquiterpene valerenic acid (C_15_H_21_O_2_^−^) [23]. Peak 15, with a [M − H]^−^ ion at *m*/*z*: 221.15488, was identified as the antifungic norsesquiterpene alcohol rishitin (C_14_H_22_O_2_^−^)[24]. Peak 19, with an [M − H]^−^ ion at *m*/*z*: 209.15430, was identified as Blumenol C (C_13_H_22_O_2_^−^, 4-(3-hydroxybutyl)-3,5,5-trimethylcyclohex-2-en-1-one) [25].

## 3. Discussion

This study, for the first time, demonstrated the hypotensive effect of the *Melicoccus bijugatus* Jacq when it is administrated orally in normotensive animals. This hypotensive effect was, in part, because of the decrease in HR. In addition, we demonstrated a partial recovery of ISO-induced myocardial infarction by the action of *M. bijugatus*.

ISO is an isopropyl analog of epinephrine that stimulates the α-1 adrenergic receptors [8], oxidative stress in cardiac myocytes, cell membrane destabilization and damage. It also increased intracellular adenylyl cyclase in the myocardium, increased lipid deposition in the myocardium, and increased the heart weight to body weight ratio and myocardial infarction [26]. ISO also significantly elevated the ST segment and shortened the QTc interval in the experimental animals; this was reversed by treatment with *M. bijugatus* extracts. The chronotropic and inotropic actions of ISO also caused an increase in the SBP. The sum of these cardiovascular changes is in an increase in HR and cardiac output and a decrease in the MAP, as observed in our study. Treatment of the animals exposed to ISO with *M. bijugatus* extracts caused decreases in HR, and MAP. This showed the cardioprotective role of the *M. bijugatus.*

This increase in the ratio of heart weight to body weight was thought to be a hypertrophic response, possibly due to an increase in the amount of protein synthesis that was occurring in the damaged tissue as it attempted to repair itself, as well as an increase the number of inflammatory cells to become mobilized in response to the damage [27]. Other possible mechanisms include increase in the glucose uptake in cardiac myocytes along with an increase in oxidative stress with ISO administration [28], accumulation of fluid in intracellular space of the tissue as well as an increase in the water content of the cells themselves [29]. Histo-morphological presentations of myocardial infarction to the ISO group were observed. There were no appreciable changes in the intima, media and adventitia layers of the aorta, caudal or vena cava (data not shown).

Using UHPLC high-resolution orbitrap mass spectrometry, (UHPLC-OT-HR-MS) we have identified 20 secondary metabolites in the aqueous extract of *M. bijugatus*, most of which, as far as we know, were reported here for the first time. Many of these compounds are simple organic acids, such as vitamin C, a flavonoid, several phenolic compounds, terpenoids, and two fatty acids. Furthermore, the results obtained in this study clearly show that the infusion can be a natural source of phenolic compounds with potential applications in the neutraceutical management of myocardial infarction.

A possible mechanism of action may be through the inhibition of the proliferation of cells of the vascular smooth muscle by caffeic acid, which is found in the fruit’s pulp tissues [30]. The hypotensive properties displayed may also be due to the action of coumaric acid derivatives. A derivative of the sugar of coumaric acid was confirmed to be a major peak in the HPLC fingerprint profile at 280nm [12]. One such derivative of coumaric acid is p-coumaric which has been known to possess antioxidant effects as well as anti-platelet activity [31].

Increases in serum levels of troponin I, CK-MB, myoglobin, and high sensitivity C-reactive protein are often associated with myocardial damage [32,33], though with some limitations [3,34]. Our study showed no significant difference in the serum concentration of CK-MB in rats induced with ISO compared to the control group (data not shown), this could be due to the timeline of the analysis of these biomarkers. Our results are supported by Zhang et al. (2008), who reported a decline in these biomarkers, down to control levels, after 48 h [35].

Myocardial infarction is often associated with an increase in the white blood cell count as part of an inflammatory response to the damaged cells that are present due to the necrosis of the tissue [36]. Other hematological features associated with myocardial infarction include an increase in whole blood viscosity and plasma viscosity, an increase in the white blood cell, leukocyte and neutrophil count [1], an increase in the erythrocyte count and hemoglobin concentration [37], and an increase in red cell indices, such as mean corpuscular hemoglobin (MCH), mean corpuscular hemoglobin concentration (MCHC) and mean corpuscular volume (MCV) [4]. Sangeetha and Quine, (2008) reported increases in hemoglobin, RBC, hematocrit, WBC and platelet counts with ISO induced myocardial infarction in male Wister rats [37], while Lobo-Fiho et al., (2011) suggested no alterations in terms of the hemoglobin indices [1]. We observed significant alterations in the hematological parameters in the treatment groups when compared to the control group. This study showed that ISO significantly reduced the white blood count (WBC), red blood count (RBC), hematocrit (HTC), mean cell volume (MCV) and mean cell hemoglobin values (MCH). The extract normalized the hematological values (WBC, RBC, HCT and MCV) depressed by ISO while significantly elevating the platelet count. The increased production of platelets suggests that the extract did not present a toxic effect, such as that of *Colocasia esculenta* (L.) Schott [38], but presented an increase in platelets similar to *Carica papaya* Linn. [39].

## 4. Materials and Methods

### 4.1. Plant Material and Extraction

The fruit of the *M. bijugatus* tree was collected in September. Mr. Patrick Lewis, Department of Botany, University of the West Indies, Mona Campus (Kingston, Jamaica), made a botanical identification of the plant (voucher specimen AN 08, 10/11). The fruit pulp was separated from the seed by hand using a knife. The pulp was then blended and the juice was squeezed out through a strainer before being filtered to remove any remaining pulp fibers. Finally, the juice was placed inside a Freeze Dry machine (Freezone 4.5 L, Labconco, Kansas, MO, USA).

### 4.2. Experimental Animals

The study was conducted according to the Animal Scientific Procedures Act of 1986 following the receipt of approval from the UWI/FMS Ethics Committee (AN 06,15/16). Twenty male Sprague- Dawley rats (8–10 weeks old and 170 g to 230 g) were chosen, housed in plastic cages at the UWI Mona campus Animal House at room temperature (22–25 °C) with a humidity of 45–51%. Fresh tap water was available to the rats via a bottle and food was administered *ad libitum*. Four groups were identified, each of which consisted of five rats. The first group served as a control group and did not receive any drugs. Rats from the second group were administered, by gavage, 100 mg/Kg of *M. bijugatus* liquid extract only daily for six weeks. The third group of rats received only two doses of 85 mg/Kg b.w. ISO intraperitoneally within a 24-hour period. Finally, rats of Group 4 were injected intraperitoneally with two doses of 85 mg/Kg b.w. of ISO within a 24-hour period. Following this, they were given 100 mg/Kg b.w. *M. bijugatus* extract, by gavage, daily for six weeks. The animals were sacrificed under anesthesia seven days after the period of treatment. The heart and kidney tissues were harvested and weighed to obtain a ratio of the heart and kidney weights to body weight. In addition, a blood sample was collected to conduct biochemical and hematological assessments among the subject groups.

### 4.3. Blood Pressure Recordings and ECG

The tail cuff method (CODA) was used to measure the SBP and DBP of the rats once before the administration of ISO and once per week following the administration of ISO. The pulse pressure (PP) was calculated using the formula PP = (SBP − DBP). The MAP was calculated using the formula: MAP = DBP + 1/3(SBP − DBP). For ECG recordings, rats were first anesthetized with ketamine (42 mg/Kg, i.p.) and xylazine (5 mg/Kg, i.p.). The ECG electrodes (BIOPAC System Inc, California, CA, USA) were placed subcutaneously in a bipolar configuration (DII). Measurements were done using the Electrocardiogram Amplifier equipment (ECG100C, BIOPAC System Inc, California, CA, USA) and tracings were recorded using the AcqKnowledge III computer software program (3.9.1., BIOPAC System Inc, California, CA, USA). The QT interval was taken as the time from the beginning of the QRS complex to the end of the T-wave. The RR interval was taken as the time elapsed between two consecutive maxima of the R-waves. The corrected QT interval (QTc) was calculated in accordance with the formula [40,41]:
QTc = QT/(RR)^1/2^

### 4.4. Biochemical Estimation

Whole blood was collected and assessed using standardized ethylenediaminetetraacetic acid (EDTA) tubes to obtain a complete blood count (CBC). The Cell Dyn Emerald Hematology Analyzer (Abbott Laboratories, Chicago, IL, USA) was then used to assess hematological parameters.

### 4.5. Histo-Morphological Appraisal

Histo-morphologic analyses of the cardiac tissues of the Sprague-Dawley rats were done using a Nikon Eclipse C*i* research microscope (Nikon Instruments Inc., New York, NY, USA). Mirco-measurements were done via integrated mechanical stages with graduated locator margins and built in slide holders, as well as X-Y translator knobs. The tissues were harvested and immediately submerged in 10% neutral buffered formalin for preservation. They were subsequently processed, embedded in wax, and serial sectioned to a thickness of 4 microns then stained with haematoxylin and eosin (H and E) stain.

Twelve (12) full-thickness transverse sections from the heart, immediately subjacent to the atrioventricular valves (three from each group), were analyzed and evaluated for edema, mononuclear cell infiltration, fibrosis, interstitial hemorrhage and myocyte degeneration. The number of foci exhibiting degenerative features was recorded and the maximum diameter of the degenerated area was measured using a stage micrometer (Nikon Instruments Inc., New York, NY, USA).

Sixteen (16) longitudinal and/or transverse sections of the large caliber abdominal blood vessels, i.e., aorta (two from each group) and caudal vena cava (two from each group), were analyzed and evaluated for evidence of endothelial injury, degeneration of the intima or elastic lamina, atherosclerotic change, or any features of adventitial cellular injury.

The Ishak systems were employed to ascribe a stage of fibrosis and to grade the degree of inflammatory changes (1). In addition to these established, semi-objective parameters, subjective evaluation was performed for sinusoidal vascular congestion and the degree of nuclear chromatin density.

### 4.6. UHPLC-DAD-MS Instrument

The use of the Scientific Dionex Ultimate 3000 UHPLC system (Thermo Fisher Scientific, Bremen, Germany) hyphenated with a Q exactive focus machine (Thermo Fisher Scientific, Waltham, MA, USA) was already reported [42]. For the analysis, 5 mg of the lyophilized material were dissolved in 2 mL of methanol, filtered (PTFE filter) and 10 µL were injected in the instrument, with all specifications set as previously reported [42].

### 4.7. LC Parameters and MS Parameters

Liquid chromatography was performed using an UHPLC C18 column (Acclaim, 150 mm × 4.6 mm ID, 2.5 µm, Thermo Fisher Scientific, Bremen, Germany) operated at 25 °C. The detection wavelengths were 254, 280, 330 and 354 nm, and DAD was recorded from 200 to 800 nm for peak characterization. Mobile phases were 1% formic aqueous solution (A) and acetonitrile (B). The gradient program time in minutes, % (B) was: (0.00, 5); (5.00, 5); (10.00, 30); (15.00, 30); (20.00, 70); (25.00, 70); (35.00, 5) and 12 min for column equilibration before each injection. The flow rate was 1.00 mL min^−1^, and the injection volume was 10 µL. The standards and resin extract dissolved in methanol were kept at 10 °C during storage in the autosampler. The HESI II and Orbitrap spectrometer parameters were optimized as previously reported [42,43].

### 4.8. Statistical Analysis

The data collected were expressed as mean ±SEM (standard error of mean). The mean values from the different groups were analyzed using GraphPad Prism Version 6.0 Software (GraphPad Software, San Diego, CA, USA). One-way or two-way ANOVA were used to compare the means, followed by the Bonferroni test. A value of *p* < 0.05 was considered statistically significant [44].

## 5. Conclusions

In conclusion, we confirmed that *Melicoccus bijugatus* partially reversed myocardial damage and injury in an experimental ISO rat model. This was indicated by changes seen in the ECGs and in the reversal of the increased heart to body weight ratio in animals with ISO induced myocardial injury and blood pressure changes associated with a normalizing or reversal of symptoms associated with cardiac injury.

## Figures and Tables

**Figure 1 molecules-24-00235-f001:**
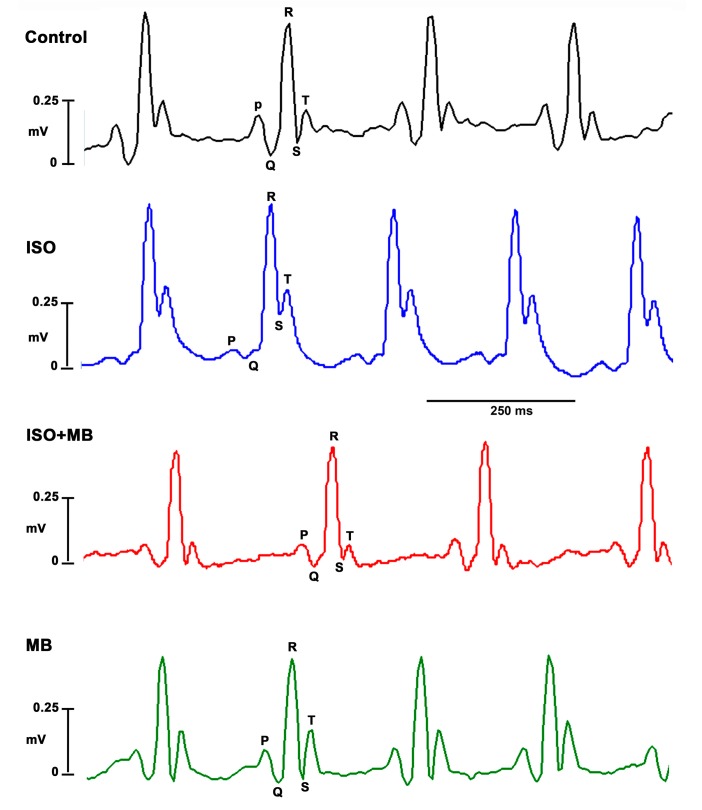
Electrocardiograms (ECG) showing the bradycardic effects of *M. bijugatus* extract (MB, 100 mg/Kg) in myocardial injury. Control shows the normal electrocardiograph. ISO (85 mg/Kg) shows an elevated ST segment; ISO + MB shows a restored ST segment.

**Figure 2 molecules-24-00235-f002:**
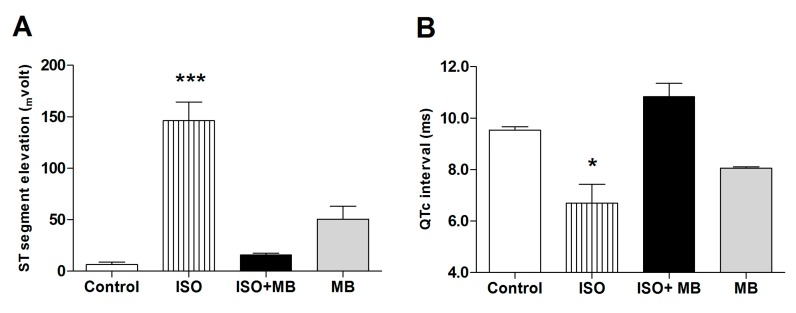
Treatment, with *M. bijugatus*, of a myocardial injury caused by ISO. The data shows the effects of MB and ISO on ST segment elevation (**A**) and the duration of the QTc interval of the ECG (**B**). ISO (85 mg/Kg) significantly elevated the ST segment (*p* < 0.001) and shortened the QTc interval (*p* < 0.05), which recovered after treatment with 100 mg/Kg of *M. bijugatus*. The extract did not, *per se*, cause any change. * *p* ˂ 0.05, *** *p* ˂ 0.001 vs. control; *n* = 5.

**Figure 3 molecules-24-00235-f003:**
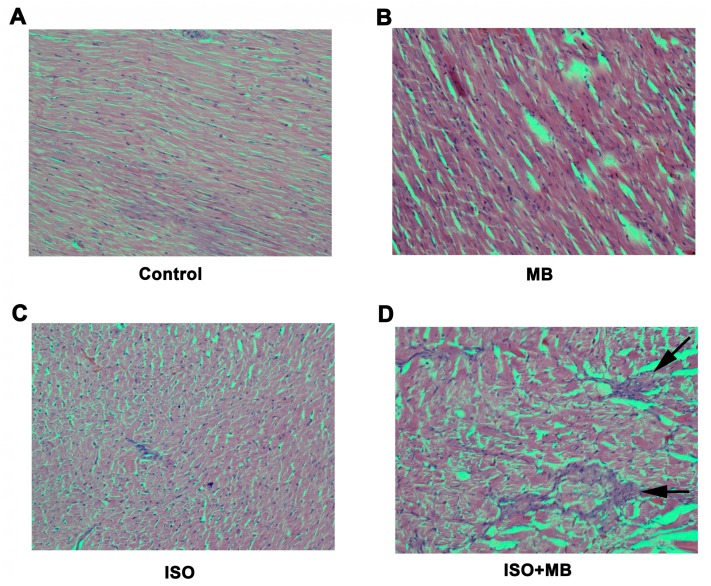
Histological analysis of myocardial injury. Histomicrograph of transverse sections of the heart [×200; hematoxylin and eosin stain] taken through the ventricles, just below the atrioventricular valves of control (**A**), 100 mg/Kg MB alone (**B**), ISO treatment alone (**C**), ISO + MB (**D**). The ISO group showed myocardial infarction in the central region of the left ventricular muscle wall (**C**). The Control (**A**) and MB only group (**B**) displayed no features of myocardial injury. The arrows demonstrated the area of fibrosis.

**Figure 4 molecules-24-00235-f004:**
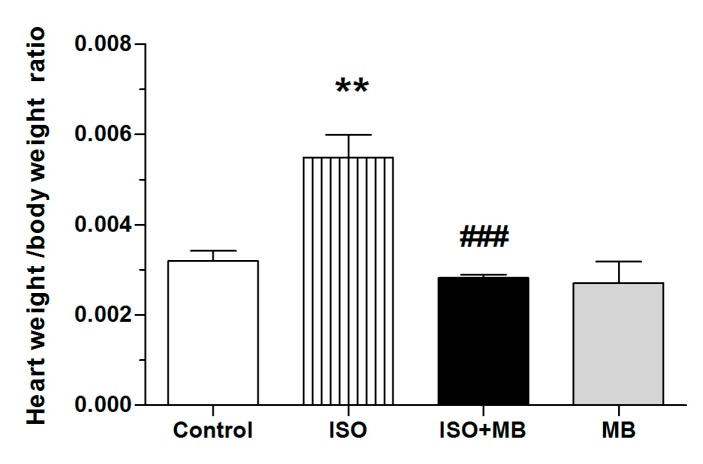
Effects of ISO and treatment with *M. bijugatus* extract on weight/body weight ratio. Depicts the ISO (85 mg/Kg) induced myocardial damage through an elevation of the heart weight to body weight ratio. This was significantly reduced in the MB (100 mg/Kg) treated groups. ** *p* ˂0.01 vs. control; ^###^
*p* ˂ 0.001 vs. ISO, *n* = 5.

**Figure 5 molecules-24-00235-f005:**
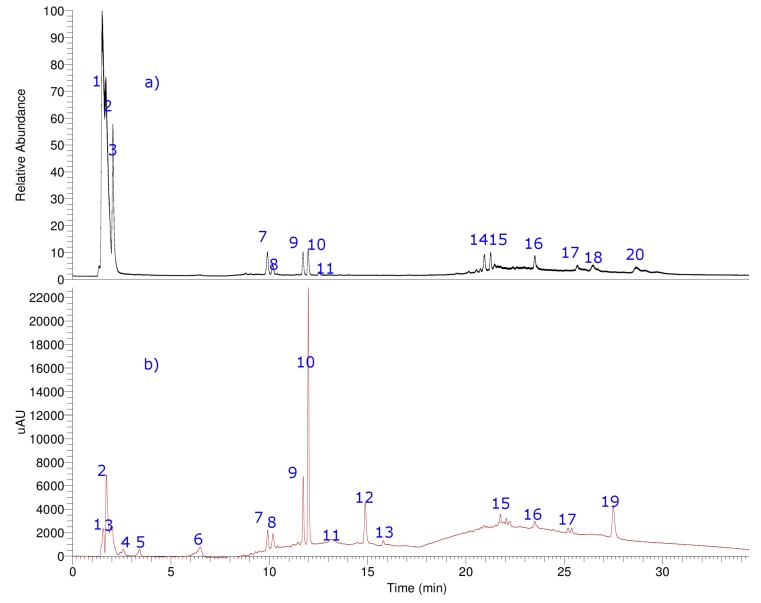
UHPLC chromatogram total ion current [total ion current (**a**), UV at 280 nm (**b**)] of aqueous extract of *M. bijugatus*. The details of metabolites are in the Appendix A.

**Table 1 molecules-24-00235-t001:** Effects of *Melicoccus bijugatus* (MB; 100 mg/Kg) on mean arterial blood pressure (MAP), systolic blood pressure (SBP), diastolic blood pressure (DBP), pulse pressure (PP), and heart rate (HR) of normotensive rats and those with myocardial damage with ISO.

	Normotensive		Myocardial Damage	
	Control	MB	ISO	ISO + MB
MAP, mmHg	101 ± 3	69 ± 2 ***	113 ± 5	83 ± 6 *^,###^
SBP, mmHg	131 ± 3	85 ± 2 ***	133 ± 4	106 ± 6 ***^,###^
DBP, mmHg	87 ± 5	60 ± 3 ***	108 ± 8 *	72 ± 8 ^###^
PP, mmHg	44 ± 5	27 ± 2 ***	28 ± 2 **	27 ± 3 **
HR, bpm	246 ± 25	146 ± 11 **	307 ± 20	181 ± 18 ^###^

Values are mean ±standard error of the mean of five experiments in mmHg. Statistically significant differences: * *p* < 0.05, ** *p* < 0.01, *** *p* < 0.001 vs. control; ^###^
*p* < 0.001 vs. ISO.

**Table 2 molecules-24-00235-t002:** Effect of *M. bijugatus* on white cell parameters in ISO-induced cardiac injury.

	Control	MB	ISO	ISO + MB
WBC (10^3^/µL)	9.5 ± 1.2	8.6 ± 1.3	4.1 ± 1.0 *	7.1 ± 1.2
LYM (10^3^/µL)	5.8 ± 0.6	5.9 ± 1.0	3.0 ± 0.8	4.9 ± 0.8
MID (10^3^/µL)	1.5 ± 0.2	1.1 ± 0.2	0.62 ± 0.2 *	1.1 ± 0.1
GRA (10^3^/µL)	2.4 ± 0.5	1.5 ± 0.3	0.5 ± 0.1 **	1.2 ± 0.3
LYM (%)	61.4 ± 2.9	66.8 ± 2.7	73.9 ± 0.9 **	69.6 ± 1.1
MID (%)	14.5 ± 0.6	13.9± 1.0	14.5 ± 0.8	13.2 ± 1.5
GRA (%)	27.3 ± 0.9	19.3 ± 2.4	11.6 ± 1.0 ***	17.3 ± 1.4 *

WBC—White blood count, LYM—Lymphocyte, MID—Others white cells, GRA—Granulocyte. Values are mean ±standard error of the mean of five experiments in mmHg. Statistically significant differences: * *p* < 0.05, ** *p* < 0.01, *** *p* < 0.001 vs. control.

**Table 3 molecules-24-00235-t003:** Effect of *M. bijugatus* on red cell and trombocytes parameters in the ISO-induced cardiac injury.

	Control	MB	ISO	ISO + MB
RBC (10^6^/µL)	6.8 ± 0.2	5.9 ± 0.4	5.2 ± 0.2 **	6.3 ± 0.42
HGB (g/dL)	14.8 ± 0.3	13.6 ± 0.6	12.5 ± 0.6	14.1 ± 1.0
HCT (%)	40.0 ± 0.8	36.2 ± 1.2	32.6 ± 1.5 **	34.8 ± 1.4
MCV (fL)	2.4 ± 0.5	1.5 ± 0.3	0.5 ± 0.1 **	1.2 ± 0.3
MCH (pg)	21.7 ± 0.2	22.2 ± 0.5	24 ± 0.6 **	22.2 ± 0.4
MCHC (g/dL)	37.1 ± 0.4	37.4 ± 0.6	38.6 ± 0.4	37.7 ± 0.2
RDW	15.0 ± 0.3	15.7 ± 0.5	15.3 ± 1.2	16.6 ± 0.9
PLT (10^3^/µL)	680.3 ± 31.0	840.0 ± 21.1 *	668.8 ± 49.0	918.8 ± 42.7 **^,##^
MPV	6.6 ± 0.2	7.7 ± 0.4	7.0 ± 0.2	7.4 ± 0.5

RBC—Red blood count, HGB—Hemoglobin, HCT—Hematocrit, MCV—Mean corpuscular volume, MCH—Mean corpuscular hemoglobin, MCHC—Mean corpuscular hemoglobin concentration, RDW—Red cell distribution width, PLT—Platelets, MPV—Mean platelet volume. Values are mean ±standard error of the mean of five experiments in mmHg. Statistically significant differences: * *p* < 0.05, ** *p* < 0.01, *** *p* < 0.001 vs. control; ^##^
*p* < 0.01 vs ISO.

**Table 4 molecules-24-00235-t004:** Identification of metabolites by UHPLC-PDA-OT-MS.

Peak	Retention Time (min)	UV Max	Tentative Identification	Elemental Composition [M − H]^−^	Theoretical Mass (*m*/*z*)	Measured Mass (*m*/*z*)	Accuracy (dppm)	MSn Ions (dppm)
1	1.72	220	Citric acid	C_6_H_7_O_7_^−^	191.01863	191.01933	3.68	
2	1.82	222	Isocitric acid	C_6_H_7_O_7_^−^	191.01863	191.01955	3.04	
3	1.65	-	Saccharose	C_12_H_21_O_11_^−^	341.10784	341.10783	2.44	
4	2.55	-	Glucose	C_6_H_11_O_6_^−^	179.05501	179.05550	2.69	
5	3.37	230	Furoic acid	C_5_H_3_O_3_^−^	111.00767	111.00787	3.25	
6	6.35	245	Salicilic acid glucoside	C_13_H_15_O_8_^−^	299.07614	299.07712	3.25	137.02440
7	9.80	245–325	Aflavarin	C_24_H_21_O_9_^−^	453.11801	453.11703	−2.1	
8	10.24	-	Trihydroxyoctadecatrienoic acid	C_18_H_29_O_5_^−^	325.20205	325.18443	−54.2	
9	11.98	275–339	Coumaric acid glucoside	C_15_H_17_O_8_^−^	325.09179	325.09277	3.01	163.0291, 145.02870, 117.03368
10	12.24	275–339	Coumaric acid galactoside	C_15_H_17_O_8_^−^	325.09179	325.09271	2.82	163.0291, 145.02870, 117.03368
11	13.35	275–339	Feruloyl glucoside	C_16_H_19_O_9_^−^	355.10346	355.10336	3.34	147.04449, 193.05058
12	15.02	254–354	Isorhamnetin-3-*O*-gglucoside	C_22_H_21_O_11_^−^	477.11679	477.11670	−27.12	314.04370
13	15.83	275–339	Feruloyl galactoside	C_16_H_19_O_9_^−^	355.10346	355.10355	3.34	147.04449, 193.05056

14	21.57	236–329	3-*O*-Caffeoylquinic acid	C_16_H_17_O_9_^−^	353.0878	353.0878	0.53	191.05608
15	21.87	-	Rishitin	C_14_H_22_O_2_^−^	221.15361	221.15488	5.74	
16	23.54	300	Embelin	C_17_H_26_O_4_^−^	293.17474	293.17587	3.87	
17	25.38	235	Sedanenolide	C_12_H_15_O_2_^−^	191.10666	191.10741	3.92	
18	26.45	225	Valerenic acid	C_15_H_21_O_2_^−^	233.15470	233.15455	3.87	149.13301
19	27.78	214	Blumenol C	C_13_H_22_O_2_^−^	209.15631	209.15430	3.30	
20	28.32	220	Hydroxyheptadecatrienoic acid	C_17_H_27_O_7_^−^	311.18640	311.16876	−56.6

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
