# Peer review of "Modulatory Effect of Guinep (Melicoccus bijugatus Jacq) Fruit Pulp Extract on Isoproterenol-Induced Myocardial Damage in Rats. Identification of Major Metabolites Using High Resolution UHPLC Q-Orbitrap Mass Spectrometry"

_molecules, 2019, doi:10.3390/molecules24020235_

Round 1
Reviewer 1 Report
This manuscript describes a hypotensive effect of Melicoccus bijugatus Jacq extract. In addition, the extract recovered myocardial damage induced by isoproterenol. The authors identified 20 ingredients in the extract by LC-q-orbitrap MS. Although the results are interesting, the authors need to address concerns shown below.
In Table 1, there is a significant difference in PP between control and ISO groups. Is this a typical symptom of myocardial damage? There is no statement about the data. The authors need to discuss the result.
The EGC of MB-treated mice is more similar to that of ISO mice than that of control mice in Figure 1. In addition, ST segment was elevated not only in ISO mice but also in MB mice, while QTc interval was shortened not only in ISO mice but also in MB mice. Based on these data, MB extract seems to induce cardiac injury like ISO. Was statistical significance found only between control and ISO in Figure 2? Is there any statistical significance between other groups?
In Table 3, is there a statistical significance in MCH between control and ISO? What about between control and MB?
In Table 3, what is the possible reason about the increase in platelet number in MB-treated mice?
P12 L298: Spell out in English.
In Table 4, please correct the decimal point of theoretical mass and measured mass.
Author Response
This manuscript describes a hypotensive effect of Melicoccus bijugatus Jacq extract. In addition, the extract recovered myocardial damage induced by isoproterenol. The authors identified 20 ingredients in the extract by LC-q-orbitrap MS. Although the results are interesting, the authors need to address concerns shown below.
In Table 1, there is a significant difference in PP between control and ISO groups. Is this a typical symptom of myocardial damage? There is no statement about the data. The authors need to discuss the result. We have tried to explain this point in the Results, section 2.1.
The EGC of MB-treated mice is more similar to that of ISO mice than that of control mice in Figure 1. We chose a new representative original ECG of the control group and, then, the Figure 1 was modified.
In addition, ST segment was elevated not only in ISO mice but also in MB mice, while QTc interval was shortened not only in ISO mice but also in MB mice. Based on these data, MB extract seems to induce cardiac injury like ISO. Was statistical significance found only between control and ISO in Figure 2? Is there any statistical significance between other groups?
In the Figure 2, the statistical analysis (one-way ANOVA, post hoc Bonferroni) indicated that ST segment was significantly (p<0.001) elevated only in ISO rat, while not significantly elevated in MB rat compared to control. QTc interval seems shortened in MB group, but it was not significantly different versus control group.
There were other statistical significance between groups, as such QTc interval significantly (p<0.01) elevated in ISO+MB group versus ISO, and QTc interval significantly (p<0.05) decreased in MB group compared to ISO+MB group, but we prefer omit this one to avoid confusion in the Figure 2B.
In Table 3, is there a statistical significance in MCH between control and ISO? What about between control and MB? Thanks for this observation. We checked the MCH values in the Table 3, and ISO value was corrected: 24±0.6 pg.
In Table 3, what is the possible reason about the increase in platelet number in MB-treated mice?
The following statement was included in the Discussion: “The increase production of platelet suggests that the extract did not present a toxic effect, such as Colocasia esculenta (L.) Schott [38], but presented an increase in platelets similar to Carica papaya Linn [39]”.
P12 L298: Spell out in English. We corrected spelling of the EDTA.
In Table 4, please correct the decimal point of theoretical mass and measured mass. We corrected all decimal points in theoretical mass and measured mass.
Reviewer 2 Report
Detailed suggestions:
1. Please add the brief descriptions of a) and b) in the caption of Figure 2.
2. In line91~92, The sentence, “There were no significant differences in the PP in the ISO only and ISO plus MB treatment groups when compared to the control.” might be wrong, because the data were significant differences. Please check it.
3. Could you also show the MS spectra of the other compounds in Supplementary material?
Author Response
1. Please add the brief descriptions of a) and b) in the caption of Figure 2.
We have included a description in the caption of Figure 2.
2. In line 91-92, the sentence, “There were no significant differences in the PP in the ISO only and ISO plus MB treatment groups when compared to the control.” might be wrong, because the data were significant differences. Please check it.
We have corrected the statement.
3. Could you also show the MS spectra of the other compounds in Supplementary material?
We have included all MS spectra of the compounds identified.